# From "Putting the Last First" to "Working with People" in Rural Development Planning: A Bibliometric Analysis of 50 Years of Research

Adolfo Cazorla-Montero and Ignacio De los Ríos-Carmenado *

Planning and Sustainable Management of Rural-Local Development (GESPLAN), Universidad Politécnica de Madrid, Avda. Puerta de Hierro, 2, 28040 Madrid, Spain; adolfo.cazorla@upm.es
* Correspondence: ignacio.delosrios@upm.es

**Abstract:** The contribution of academics and researchers to the discussion around sustainable rural development planning and its impact on rural communities has grown exponentially in recent years. Understanding trends in sustainable rural development research requires considering the different factors involved and affecting people from a holistic approach. This review examines, through bibliometric studies, the scientific knowledge generated on sustainable rural development planning in the last 50 years, analysing 6895 articles published in journals between 1970 and 2020. The results reveal the existence of three clusters, and important growth is observed to respond to the continuous needs in relation to sustainable rural development. This research shows the evolution of a new approach for the planning of sustainable rural development projects in postmodernity: Working with People (WWP). This WWP model, as a conceptual framework from social learning, has been validated as a novel proposal in numerous contexts. The bibliometric analysis shows an evolution in "From Putting the Last First" to "Working with People in Rural Development" research and the contributions of influential teachers, such as Chambers and Cernea. These bibliometric analyses demonstrate the correct approach of the WWP model and open new fields of research in the planning of sustainable rural development projects.

**Keywords:** rural development; sustainable development; planning; working with people

## 1. Introduction

The concept of rural development has been evolving significantly since the mid-1960s until today, moving away from a traditional approach based on the idea of modernisation, according to which all societies evolve in a linear manner from a non-rational and technologically limited state to a rational and technologically advanced state. This transition represents a move from a traditional society to a modern one [1].

Since the initial development strategies, in approximately 1965, some French rural areas linked the concept of development with planning, in the sense of driving policies of a centralist character and with a modernising spirit.

It was in the 1970s and 1980s when this idea of rural development started to evolve towards a more local perspective centred around people, such that this concept which was previously associated with economic growth and modernisation, started to gain a qualitative dimension, which started to value the quality and sustainability of growth [2].

This change was accompanied by regional planning, which in Europe especially represented a bottom-up approach, with new approaches which started to replace the traditional top-down approach from the previous decades.

The 1990s marked a definitive leap in terms of the aforementioned, with what is known as the LEADER community initiative in Europe. This represents the structural birth of an endogenous planning approach with a new way of thinking which emerges from the decline of the so-called modern project and the arrival of postmodernity [3,4],

replacing sectoral approaches with territorial rural development, which involves respect for the environment and achieving sustainability.

The planning that took place from the 1990s until the present day in Europe as a result of this Leader approach has not had a methodological comparison in other parts of the world. It is since the second decade of the 21st century that new methodological approaches appeared, reflected by more modern methods than those that are being developed in the 27 European countries and which consider people as the central part of sustainable development with planning that is adequate for this new step forward [5]. One of these new models is coined as Working with People WWP and is shown in Section 2.

Elsewhere, the growth of powerful tools for data processing now enables access to the databases of scientific publications and quantitative research, which has previously been described briefly. Within the multitude of databases for analysing the evolution of the rural development concept based on its authors and the influence of these at a global level, the methodology implemented by the CSIC Cybermetrics Lab [6,7] introduces a new approach to classifying universities. This CSIC (Cybermetrics Lab) Web ranking is supported by Google Scholar (GS) and Scopus (Elsevier, Amsterdam, The Netherlands).

Scopus has more than 30,000 indexed records and enables the quick and transparent analysis of excellence in research [8] and is used for bibliometric analysis. Google Scholar (GS) shows the level of impact based on the number of citations for each publication, and although it is not reflected in impact records and therefore lacks the quality control of many publications, it is a great help for disseminating information and analysing the impact that the most influential authors can have on other researchers. In many countries, the circulation of this relevant study, which reaches thousands of people, is not always covered by indexed journals. The highly cited authors generally have profiles on Google Scholar and other institution websites or social platforms, which means the relative impact of these influential authors can be estimated as GS is an alternative or complementary resource to the leading databases [9]. Furthermore, studies show that in all areas of knowledge, Google Scholar (GS) citations are a superset of WoS and Scopus, representing substantial additional coverage that is of interest [10]. GS finds a significantly higher number of citations than WoS and Scopus in all subject areas, as it includes many other documents. Approximately half (between 48–65% depending on the subject) of the GS citations are from journal articles, and the other half of the documents are doctoral theses or master's theses (in universities' repositories), books or chapters from books, conference proceedings, unpublished materials (such as pre-prints) and other types of documents.

Studies [8,10–12] confirm very solid correlations between GS, WoS and Scopus in all categories, despite the greater number of additional citations found in GS. This information is, therefore, very interesting for evaluating the impact of the research [12,13].

Furthermore, the GS citations are particularly useful when there are reasons to believe that the documents not covered by WoS or Scopus are important for evaluation [10], as is the case for rural development. Therefore, in the three development stages of the rural development concept, the bibliometric analysis carried out using the publications indexed by Scopus was complemented to a level of excellence by the Web of Science (WoS). A complementary analysis was provided by Google Scholar (GS) to show the evolution and dissemination of the knowledge acquired from the most influential authors on rural development in the last 50 years.

## 2. Working with People in Rural Development Projects

WWP is coined with the expression Working with People and is understood as the professional practice of rural development as a team that seeks to connect knowledge and action for a common project, which in addition to the technical value of production—of goods and services—mainly incorporates the value of the people who get involved, participate and develop through the actions carried out in the context of the sustainable project [5]. The expression Working with People is intended to show the need to overcome the technical vision of a rural project, focusing on individuals' behaviour and the context in

which they work and value beyond the project's sophistication, the improvement of human behaviour achieved by the involved agents.

With this important human dimension, the WWP model includes the following principles and values: (a) Respect and primacy for the people, which are the main elements to be considered in any development strategy and in the design of any technical innovation; (b) To guarantee social well being and sustainable development of rural communities; (c) Bottom-up and multidisciplinary approach to guarantee a subsidiary principle, in which rural development projects are the responsibility of rural community agents, considering representative actors from the different activity areas developed among them; (d) Endogenous and integrated approach which will take into account all the aspects which will allow creating new combinations and synergies generating new projects and new activities, with the intervention of socio-economic agents and managers through plurisectorial interventions.

In addition to the above principles, the WWP approach may be summarised around three dimensions—ethical–social, technical–entrepreneurial and political–contextual—which interact through social-learning processes. These three components include the four areas of the social-relations system—political field, public administration field, private and entrepreneurial fields, and Civil Society field—as a synthesis of the social model.

The apparent simplicity of the WWP project involves a large complexity given by the richness of relationships and learning that occur between the three types of agents of the proposed model (Figure 1), where the three components must be present in any project designed from the WWP approach, interactions and overlaps between them through social learning processes. WWP model requires adequate social integration from the beginning to "bring closer" potentially affected people to work with them. The greater the social complexity and the more diverse the expectations of the parties involved are, the more a sophisticated integration approach is required to make the WWP project behave as an open system, capable of entering into "dialogue" and working "with" people. This integration process exceeds simple participation and requires time to develop the ability to "listen" and to look for shared responsibility [5].

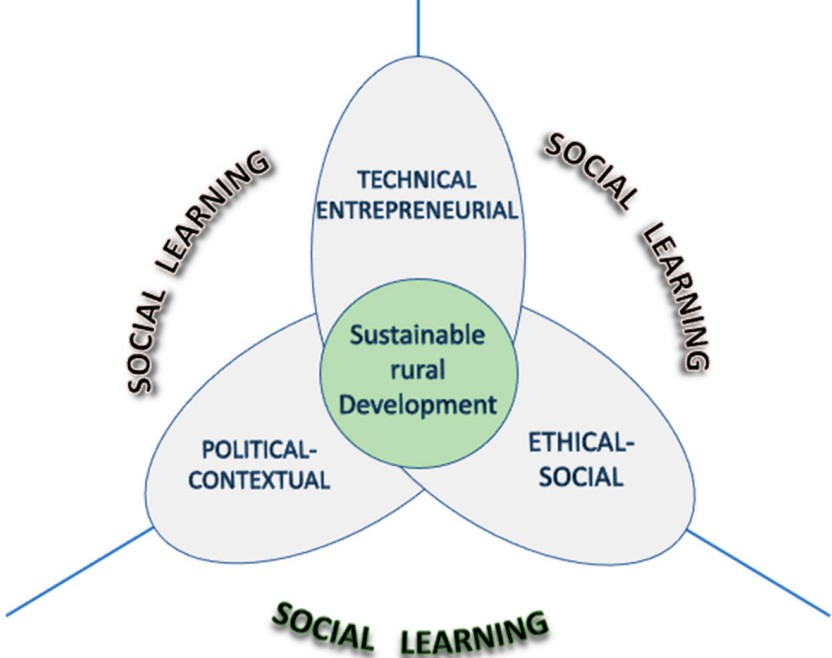

**Figure 1.** WWP dimensions [5].

The ethical–social component covers the context of behaviour, attitudes and values of people who interact to promote and manage sustainable rural development projects.

This component is identified with the social subsystem, consisting of all interpersonal relationships that are taking place within society, and it sets out the "foundations" to make people, both from private and public fields, come to work together with commitment, confidence and personal freedom.

The technical–entrepreneurial component integrates the key elements to achieve providing a sustainable rural development project as an investment unit and a technical tool capable of generating a flow of goods and services and meeting some targets according to requirements and quality standards. From the point of view of social relations, this component corresponds to the private-entrepreneurial field. The goods and services from the WWP model are the results of a "dialogue" of the agents, inserting sensitivities and being able to express emotions, cultural values, and historical references . . . reaching their maximum development when people are able to put sense in what they produce and create.

The political–contextual component covers the ability of rural development projects to make relations with political organisations and with the different public administrations. This ability to make relations with the context depends on the acquisition of an internal organisation for the project, which facilitates participation and social dynamisation. The configuration of sustainable rural development projects must ensure that organisational change processes and structural processes are generated to allow adaptation to the priorities of involved people, also working with actors from the political and public administration fields. WWP organisation has, therefore, an instrumental character in serving the population, and it is flexible and changing according to the learning and the new information generated. This way, the WWP organisation becomes a living entity that transmits values to society—from its ethical component—and is capable of influencing and changing political priorities and working together.

Finally, the social learning component provides the WWP project with an integrating component to ensure space and social learning processes among the different subsystems, which leads to learning from the real agents of change. The integration of various knowledge sources and learning forms comes to the forefront of the WWP model as a key aspect in surviving, adapting, developing and prospering in rural areas. The social learning process runs with the main assumption that all effective learning comes from the experience of real change. The population affected by the project actively participates in planning, with their own behaviours, attitudes and values—ethical–social component—to promote, manage and direct the WWP project. Therefore, the WWP approach requires generating actions directed to integrate the experienced knowledge of the affected population, along with the planner's expertise, providing mutual learning. This approach confirms that informal knowledge generated in local contexts tends to be holistic as it considers the complexity of the realities in rural areas and integrates many or at least several of the environmental, economic, social, financial, technical, political and other dimensions into a single whole. To ensure these social learning areas and processes, it is required to have a proper appreciation of values, defined as the ability to understand the inherent qualities of others and understand their points of view. This leads us again to say that the ethics and behaviour of the people involved are the basis of WWP.

This Working with People (WWP) model—from planning as social learning and from the new postmodern sensibility—has been applied in several experiences in rural development projects. As will be seen in Section 3, these experiences provide evidence that the process of social learning in rural development projects can be effective for different initiatives in the public and private sectors. The more recent engagement of informal and formal knowledge in multi-actor knowledge networks and closer collaboration with different social agents indicates the development of more participatory, inclusive and comprehensive knowledge and learning processes.

This paper examines, through bibliometric studies, the scientific knowledge generated on sustainable rural development planning in the last 50 years and shows the evolution of the WWP approach as a conceptual framework from social learning has been validated as a novel proposal in numerous contexts.

Especially the analysis shows an evolution "From Putting the Last First" to "Working with People" in rural development research and the contributions of influential authors, such as Chambers and Cernea.

## 3. Materials and Methods

A bibliometric analysis is carried out as a widespread and precise technique for examining large volumes of scientific data, understanding the interconnection between subjects and showing the current situation regarding a research topic based on different analyses: citation, co-citation and co-authorship [14].

The collection of data for bibliometric analysis was carried out using internationally recognised digital platforms which offer high-quality standards (Scopus & WoS) [12] and are the main tools used in this analysis [13]. The articles were selected using keywords, which is an established way of effectively analysing knowledge and getting a general understanding of the study [12] within the period 1970–2022. The following steps were followed:

**Phase 1**: Firstly, the publications were extracted based on the first group of subject keywords (Step 1 in Figure 1) from the Scopus database in September 2022. The search criteria were based on the "rural development" keyword featured in the title itself, the abstract or the keywords within the scientific texts. During this phase, 106,472 documents were discovered. The "planning" keyword was used as a second filter, selecting 6894 indexed documents which contained the two related terms "rural development planning". This first set of articles was published between 1970 and 2022, as previously explained. Furthermore, using the same group of keywords, the data were compiled separately for Web of Science (WoS), obtaining a second group of 1031 articles within the same period between 1970 and 2022. Finally, citation data from Google Scholar (GS) were used as a way to complement the analysis and identify the most relevant documents. For this citation data from Google Scholar, the free software "Publish or Perish" [14] was used as a practical way of extracting more data from GS and complementing the analysis [8].

**Phase 2**: The bibliographic records were downloaded for the searches so that they could be analysed for each of the periods between 1970 and 2022, as previously explained: 634 publications in period 1 (1970–1989), 1889 publications in period 2 (1990–2009) and 4371 publications in period 3 (2010–2022). Names and affiliations, titles, keyword categories and lists of references were downloaded.

**Phase 3**: To improve the credibility and validity of the study's results, data cleansing and refinement were carried out. The number of citations in the research articles was the criteria for improving the reliability of the results, which is a usual practice for data cleansing in this type of study [15]. Articles with either a single or no citations at all are eliminated because of the low impact they have on establishing the intellectual roots of the field of study.

**Phase 4**: In the following phase, keyword co-occurrence analysis and clustering were carried out [16,17]. From the overall base of 6894 indexed documents, this co-occurrence analysis identified the main research topics and trends in the area of rural development planning. Furthermore, scientific mapping was performed for a spatial representation of the relationships [18].

VOSviewer [19,20] software version 1.6.7 was used for the bibliometric analysis, which can import original databases from the ISI WoS and Scopus in a CSV format in order to visualise and analyse trends. With the help of the software, the different bibliometric techniques used in the analysis of bibliometric performance for analysing the frequency of words and citations [10], analysis of co-occurrence and clustering of keywords [16,17], and scientific mapping [18] were combined. This process of summarising and cleansing the data is described in Figure 2, and the search results for each step are shown in the table.

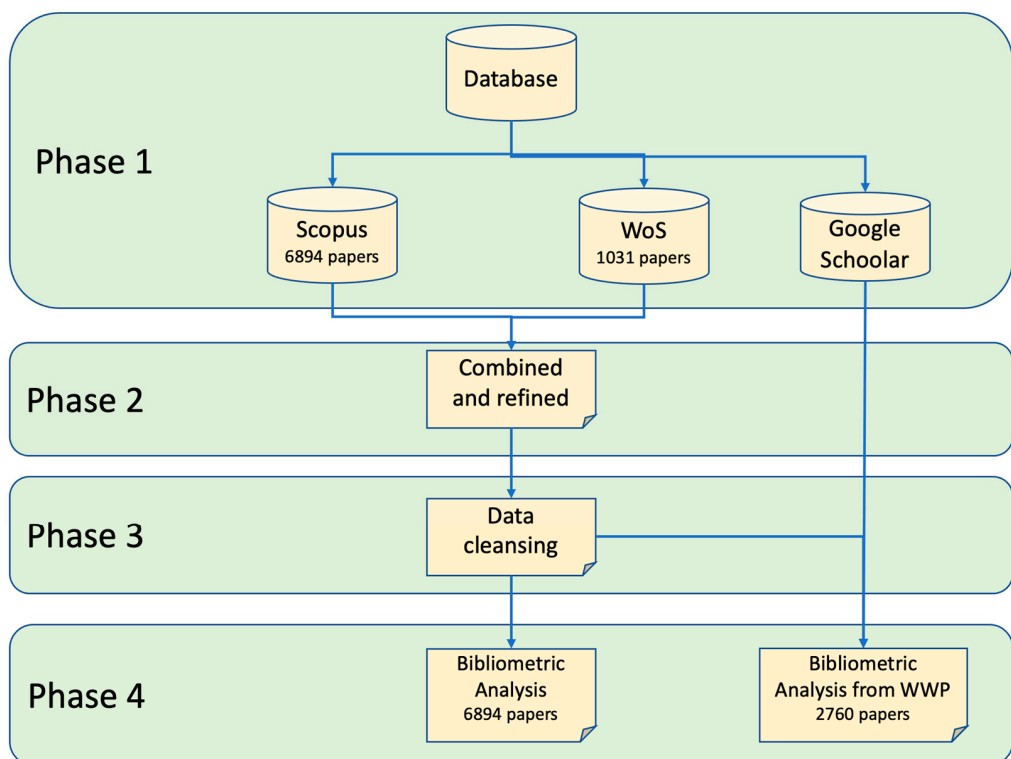

**Figure 2.** Phases of the analysis process.

## 4. Results of the Bibliometric Analysis

In this section, we provide a bibliometric analysis of the publications related to the evolution of rural development. The distribution of publications and citations, the sources that cite the publications and the most cited articles are presented in the first section at a global level. Section 4.1 shows the evolution of the most active articles and journals, Section 4.2 shows the most influential authors, and Section 4.3 includes an analysis of the co-occurrence and clustering of keywords. Finally, in Section 4.4, the bibliometric analysis is complemented by one of the most advanced models, "Working with People".

### 4.1. General Indicators for Activity and Scientific Publications

The study has identified 6894 articles published in 1512 journals between 1970 and 2022, which contain the keywords "planning" and "rural development". An ongoing increase in the number of articles during the analysed period between 1970–2022 is observed (Figure 3). Specifically, in the third period (2010–2023), the quantity of articles published on the subject has seen significant exponential growth, which shows the growing interest and intense scientific debate at an international level regarding rural development planning.

These numbers confirm how research in this field has had a growing impact, especially in the last decade between 2010–2022 (Figure 4, Table 1). The analysis also shows a great diversity of journals of impact, rather than being limited to a few journals, which demonstrates the interdisciplinary focus of rural development planning. The data show that 160 journals contain 50% of the publications. The most cited journal is Land Use Policy, followed by Landscape and Urban Planning and Journal of Rural Studies. Table 2 shows the journals ordered according to the total % of citations received (TC). Land Use Policy, Sustainability & Journal of Rural Studies stand out based on the number of articles. These data demonstrate that these journals play an especially important role within the general debate on rural development planning, although there are many others that are also highly multidisciplinary and of great importance. The key ones to highlight are World Development, Sociologia Ruralis, Journal of Cleaner Production, Science of the Total

Environment, Habitat International, Journal of Environmental Management, Community Development Journal, and European Planning Studies.

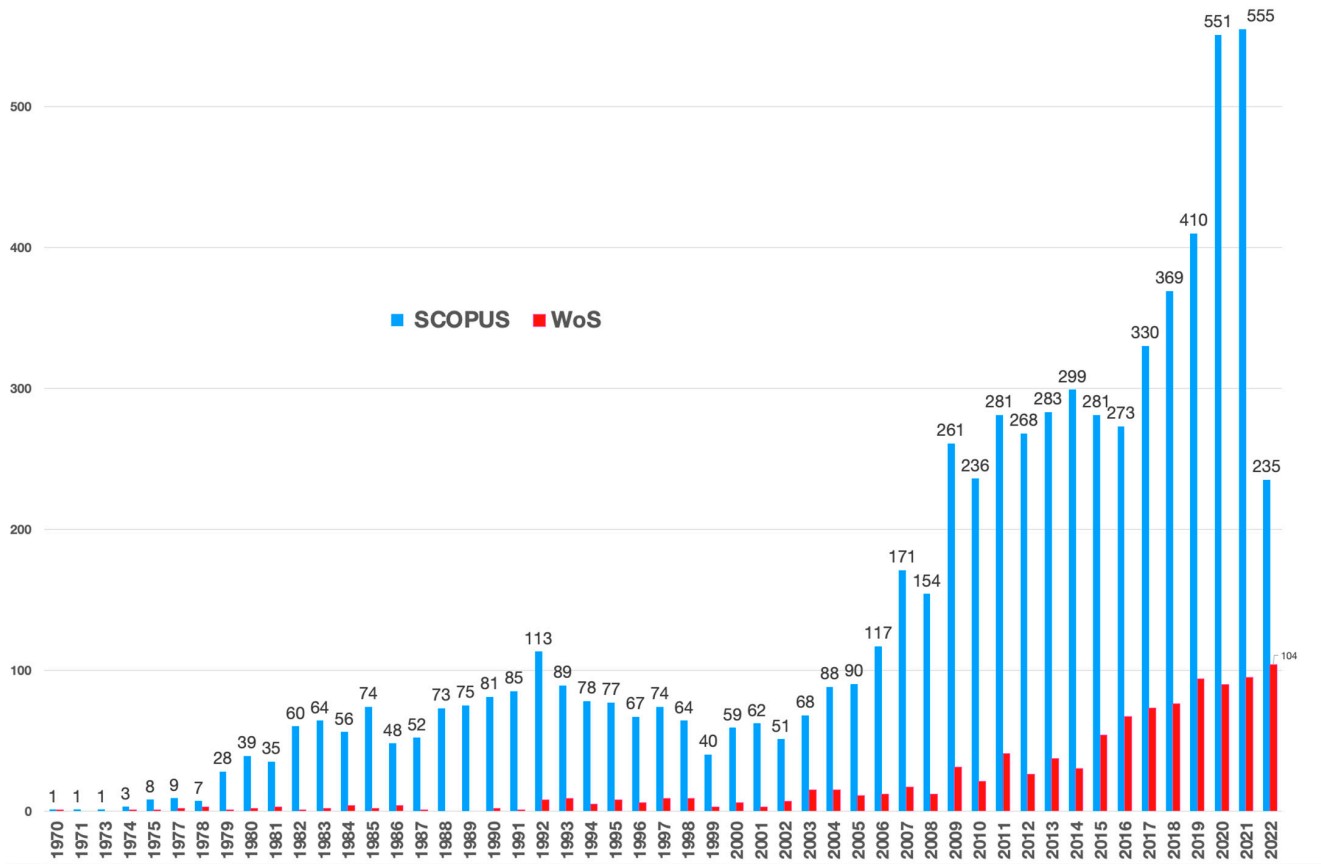

**Figure 3.** Distribution of Scopus and WoS publications by year from 1970–2022 (2 March).

**Table 1.** Search results by steps.

|  | 1st Selection Scopus | 2nd Selection WoS | 3rd Data Cleansing | 4th Bibliometric Analysis |
|---|---|---|---|---|
| Rural development and planning | 6894 | 1031 | 5522 | 5522 |

Selection of papers was made from keywords.

**Table 2.** Top 25 most cited journals Scopus y WoS.

| Journal | WoS | | Scopus | | Total | |
|---|---|---|---|---|---|---|
|  | TP [1] | TC | TP | TC | TC | % TC |
| Land Use Policy | 52 | 1622 | 125 | 4756 | 6378 | 17.9% |
| Landscape and Urban Planning | 14 | 1419 | 70 | 4739 | 6158 | 17.3% |
| Journal of Rural Studies | 23 | 715 | 125 | 3657 | 4372 | 12.3% |
| World Development | 5 | 523 | 20 | 2630 | 3153 | 8.9% |
| Sociologia Ruralis | 8 | 619 | 25 | 1372 | 1991 | 5.6% |
| Journal of Cleaner Production | 7 | 200 | 43 | 1311 | 1511 | 4.2% |
| Science of the total Environment | 5 | 125 | 47 | 1378 | 1503 | 4.2% |
| Journal of Geographical Sciences | 8 | 410 | 50 | 1034 | 1444 | 4.1% |
| Sustainability | 48 | 277 | 113 | 1098 | 1375 | 3.9% |

**Table 2.** *Cont.*

| Journal | WoS | | Scopus | | Total | |
|---|---|---|---|---|---|---|
| | TP [1] | TC | TP | TC | TC | % TC |
| Habitat International | 7 | 161 | 28 | 899 | 1060 | 3.0% |
| Journal of Environmental Management | 8 | 264 | 25 | 647 | 911 | 2.6% |
| Applied Geography | 5 | 130 | 12 | 718 | 848 | 2.4% |
| Agricultural Systems | 5 | 174 | 13 | 498 | 672 | 1.9% |
| Biomass & Bioenergy | 6 | 134 | 25 | 529 | 663 | 1.9% |
| Geoforum | 5 | 132 | 16 | 524 | 656 | 1.8% |
| Regional Environmental Change | 6 | 113 | 3 | 495 | 608 | 1.7% |
| European Planning Studies | 6 | 113 | 21 | 332 | 445 | 1.3% |
| International Regional Science Review | 9 | 61 | 19 | 331 | 392 | 1.1% |
| Mountain Research and Development | 7 | 170 | 17 | 195 | 365 | 1.0% |
| I.J. Of Environmental R. & Public Health | 5 | 31 | 34 | 245 | 276 | 0.8% |
| Computers and Electronics In Agriculture | 8 | 93 | 12 | 160 | 253 | 0.7% |
| Cuadernos de Desarrollo Rural | 6 | 91 | 6 | 68 | 159 | 0.4% |
| European Countryside | 11 | 58 | 19 | 100 | 158 | 0.4% |
| Land | 28 | 114 | 4 | 10 | 124 | 0.3% |
| Third World Planning Review | 4 | 19 | 8 | 73 | 92 | 0.3% |

[1] TP: Total publications; TC: Total citations.

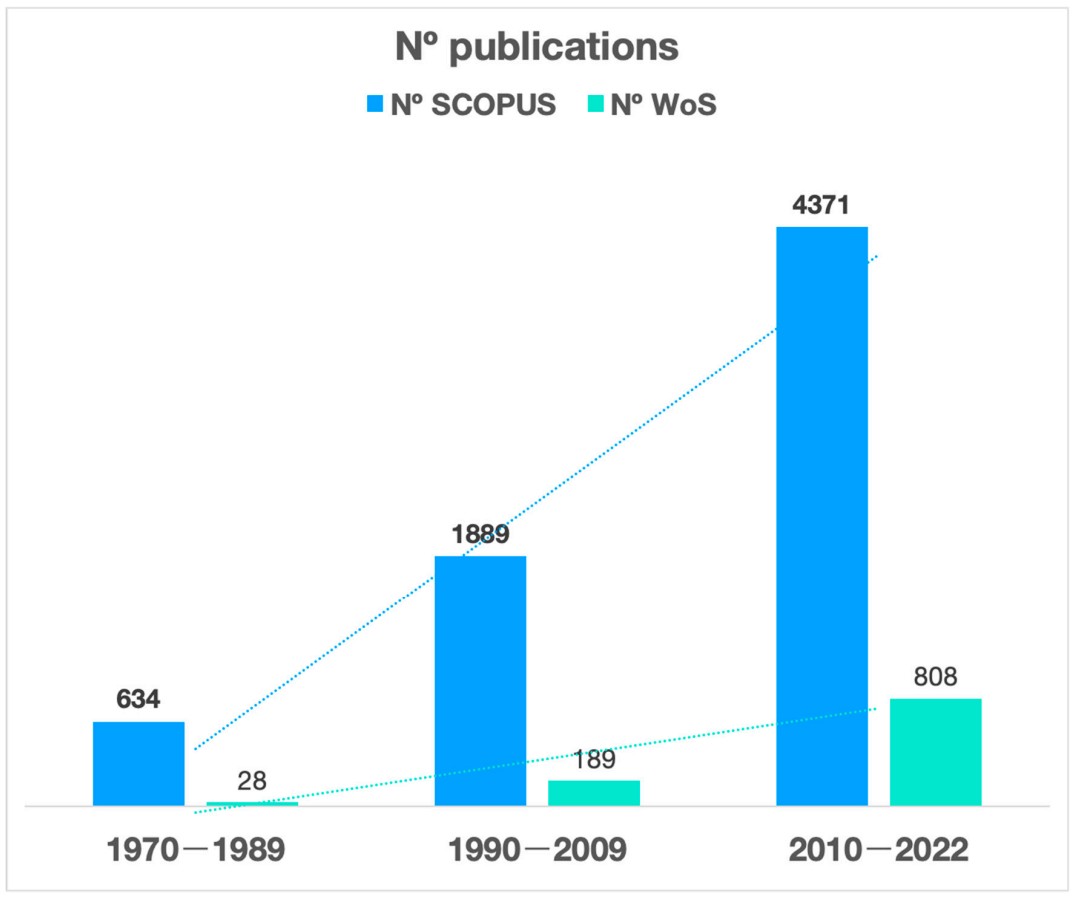

**Figure 4.** Distribution of Scopus and WoS publications by historical stages (2 March).

It is evident that the articles play a central role in the scientific debate on rural development planning, with the weight of the total number of citations (TC) being very high (83%). However, there are cases of books (Table 3) from relevant authors that are not indexed but are heavily cited and are highly influential as they are studies within Google Scholar (GS), providing substantial additional coverage to the publications in WoS and Scopus.

**Table 3.** Document type: Total publications and citations.

| Document Type | TP [1] | % TP | TC [1] | % TC |
|---|---|---|---|---|
| Article | 4645 | 68.45% | 78190 | 83.49% |
| Book | 16 | 0.24% | 2250 | 2.40% |
| Book Chapter | 70 | 1.03% | 174 | 0.19% |
| Conference Paper | 1813 | 26.70% | 4517 | 4.82% |
| Review | 242 | 3.57% | 8516 | 9.09% |
| Total, general | 6786 | 100.00% | 93647 | 100.00% |

[1] TP: Total publications; TC: Total citations.

### 4.2. Analysis of Influential Authors by Periods

This sub-section presents an analysis based on historical periods. The most influential articles, journals and authors are analysed for each of these.

#### 4.2.1. First Period 1970–1989: Introduction and the First Influential Authors

For this first period, during the process of selecting the "planning" and "rural development" keywords, 634 articles published in Scopus and 28 articles in JCR were identified. In this period, the researchers highlight the problems encountered in terms of planning and implementing rural development projects in the 1960s, typically of a large scale and with external funding from the experiences of USAID and the World Bank [21]. During the 1960s, in Europe and the United States, processes for involving the inhabitants of rural areas were not considered in development planning. Considered as being backwards, the rural regions were seen as being incapable of developing on their own [22,23].

These projects are implemented based on a traditional approach in terms of the rural development concept, relating to the idea of modernisation [1]. It was in the 1970s and 1980s that this idea of rural development started to evolve to a more local perspective centred around people, which meant that the concept that was previously associated with economic growth and modernisation started to gain a qualitative dimension that placed greater value on the quality and sustainability of growth [2]. In the 80s and 90s, numerous debates emerge regarding the term "endogenous development", especially in Europe [24,25], recognising the importance of local participation and the creation of new local organisation structures [26] in the development process.

The first works [27,28] related to Integrated Rural Development, therefore, emerged as a new planning concept which emphasises the need for an integrated approach and the need for greater participation in the design of development programmes. They start to talk about ideas regarding the mobilisation of people and taking into account the needs of diverse social groups, as well as establishing links between them [29]. However, in this phase, the studies are limited to theories and strategic considerations for the implementation of rural development policies [30,31].

To help planners and managers implement the first development strategies based on centralist and modernising policies, tools for monitoring and evaluating projects were developed, some of which are widely used, such as the Logical Framework Approach [32].

The concept is also enriched by so-called community development as a form of intentional, planned and targeted change relating to theories of social change, as well as decision-making by the rural community itself [33,34]. Other researchers put the emphasis on people, highlighting the importance of "learning from experience" [35] as a way of

improving the effectiveness of rural development projects and programmes. These authors show the limitations of the conventional, technical and quantitative models and start to consider the need for alternative approaches as useful avenues for rural development [29,34,36]. Table 4 shows the citations of these most influential references.

**Table 4.** Documents published and a number of citations (period 1970–1989).

| Authors | Title | Year | Source Title | Cited by |
|---|---|---|---|---|
| Coleman, G. | Logical framework approach to the monitoring and evaluation of agricultural and rural development projects | 1987 | Project Appraisal | 63 |
| Hulme, D. | Learning and not learning from experience in rural project planning | 1989 | Public Administration and Development | 34 |
| Anyanwu, C.N. | The technique of participatory research in community development | 1988 | Community Development Journal | 22 |
| Morss, E.R., Gow, D.D. | Implementing rural development projects: lessons from AID and World Bank experiences | 1985 | World Bank experiences | 13 |
| Livingstone, I. | On the concept of 'integrated rural development planning' in less developed countries | 1979 | Journal of Agricultural Economics | 12 |
| Leupolt, M. | Integrated rural development: key elements of an integrated rural development strategy | 1977 | Sociologia Ruralis | 11 |

4.2.2. Second Period 1990–2009: Transition Based on the Human Dimension

Within this period, 1889 articles published in Scopus were identified, of which 189 articles are JCR which contain the "planning" and "rural development" keywords. In the 1970s, it was believed that rural development projects were at the forefront of initiatives to improve rural livelihoods. However, subsequent evaluations and studies showed rural development projects in a bad light [37]. In response to the poor results, this second period saw an extensive debate on how to improve rural development planning with new approaches and methodologies for working with communities and promoting people's participation.

Robert Chambers is one of the most influential authors, and based on his extensive intellectual and practical work, he considered the need for rural development planners to take on a more humble role, listening and learning from the population [38]. He appeals to the scientific community, development professionals and policymakers from an ethical and practical point of view [39], explaining the mistakes in development practice and calling for changes in learning methods, behaviours and values to prioritise people, especially the poor, weak and vulnerable [40,41]. His call for participation has materialised in practical methodologies, which have been used by researchers and professionals across the world (Musyoki, 2022), with his famous "Participatory Rural Appraisal" (PRA) [42] and "Putting the first last" [43] slogan standing out. Ian Scoones, a student of Chambers from the same Institute of Development Studies at the University of Sussex, is another influential author from this period, with methodologies that integrate the rural population's knowledge in the planning process [44,45].

Cernea is another important author, and since the early 1990s, his work "Putting people first: sociological variables in rural development" has addressed the adequacy and entry points of sociological knowledge in the planning of development projects. His work presents new emerging approaches for integrating sociological knowledge in the design and implementation of development programmes and projects [46], opening up a new field of research based on the social components of sustainability. With his "Putting people

first" slogan, he highlights the lack of recognition for the role of "social actors" within sustainability, condemning environmental problems due to the lack of consideration of human aspects rather than because of economic or technical factors. This is, therefore, a precursor for considering sustainability based on the three dimensions (social, economic, ecological) [47], along with other planners who incorporate the notion of care and respect based on an integrated vision [48].

In his work "People First", Burkey [49] complements Chambers' work by developing new participative methodologies for rural development planning and implementation, incorporating new principles, such as sustainability, awareness, local control, cooperation and autonomy. The approaches relating to endogenous development, theories on innovation and social learning involving networks of actors in different rural contexts [50] are enriched at this stage. Other authors developed participatory methodologies applied to rural development [49] and incorporated the so-called social learning [51–53] to overcome social conflicts in the planning of sustainable rural development using new skills demanded by planners [39,54,55].

However, the big change in Europe in the 1990s was driven by the Leader community initiative, combining the birth of an endogenous planning approach with a new way of thinking born out of the decline of the so-called modern project and the arrival of postmodernity [56]. These influential authors, along with many others [57–60], contribute to an expansion of sustainable rural development, as an area of great interest for professionals, managers and researchers, with a broad discourse that integrates areas of knowledge and seeks alternative paths based on methodologies and practices [61].

Sectoral approaches are being replaced by territorial rural development, which entails respect for the environment and the search for territorial balance. The approaches and ways of life-based on sustainability are beginning to be seen as new ways of moving towards rural development based on an intersectoral and multidisciplinary approach [62,63]. In this second period, this new approach to planning in the European Union develops a culture of evaluation of rural planning, especially in the context of the LEADER initiative, based on methodologies aimed at developing skills and empowering the population [64], generating social learning [59], improving governance and the sustainability of rural development [65,66].

In general, the research in this period concludes that, although there is not a single model for planning rural development projects, planning through dialogue [67] and social learning [59] represent a major challenge for the renewal of models and shaping new bottom-up development trajectories. In the early 1990s, few would have anticipated the expansion of applied social science and the recognition it would have received in rural development planning [46]. The most influential references and their citations are shown in Table 5.

**Table 5.** Documents published and a number of citations (period 1990–2009).

| Authors | Title | Year | Source Title | Cited by |
|---|---|---|---|---|
| Chambers, R. | Whose reality counts? Putting the first last | 1997 | Whose reality counts? Putting the first last | 2029 |
| Chambers, R. | The origins and practice of participatory rural appraisal | 1994 | World Development | 1376 |
| Murdoch, J. | Networks—A new paradigm of rural development? | 2000 | Journal of Rural Studies | 473 |
| Renting, H, Rossing, W.A.H., Groot, J.C.J., Van der Ploeg, J.D. | Exploring multifunctional agriculture. A review of conceptual approaches and prospects for an integrative transitional framework | 2009 | Journal of Environmental Management | 341 |
| Brandon, K.E., Wells, M. | Planning for people and parks: Design dilemmas | 1992 | World Development | 304 |

**Table 5.** *Cont.*

| Authors | Title | Year | Source Title | Cited by |
|---|---|---|---|---|
| Ellis, F., Biggs, S. | Evolving themes in rural development 1950s–2000s | 2001 | Development Policy Review | 254 |
| Leeuwis, C. | Reconceptualising participation for sustainable rural Development: Towards a negotiation approach | 2000 | Development and Change | 226 |
| Burkey, S. | People first: a guide to self-reliant participatory rural development | 1993 | People first: a guide to self-reliant participatory rural development | 224 |
| Cernea, M.M. | Putting people first: sociological variables in rural development. Second edition | 1991 | Putting people first: sociological variables in rural development. | 132 |
| High, C., Nemes, G. | Social learning in LEADER: Exogenous, endogenous and hybrid evaluation in rural development | 2007 | Sociologia Ruralis | 111 |
| Ray, C. | Towards a meta-framework of endogenous development: Repertoires, paths, democracy and rights | 1999 | Sociologia Ruralis | 100 |
| Bruckmeier, K. | LEADER in Germany and the discourse of autonomous regional development | 2000 | Sociologia Ruralis | 63 |
| Barke, M., Newton, M. | The EU LEADER initiative and endogenous rural development: The application, of the programme in two rural areas of Andalusia, Southern Spain | 1997 | Journal of Rural Studies | 59 |
| Perez, J.E. | The LEADER programme and the rise of rural development in Spain | 2000 | Sociologia Ruralis | 56 |
| Cazorla, A. De los Ríos, I &. Díaz-Puente, J.M. | The LEADER community initiative as rural development model: Application in the capital region of Spain | 2005 | Agrociencia | 37 |
| Diaz-Puente, J.M., Yage, J.L., Afonso, A. | Building evaluation capacity in Spain: A case study of rural development and empowerment in the European union | 2008 | Evaluation Review | 26 |
| Marsden, T., Bristow, G. | Progressing integrated rural development: A framework for assessing the integrative potential of sectoral policies | 2000 | Regional Studies | 21 |
| Hulme, D. | Projects, politics and professionals: Alternative approaches for project identification and project planning | 1995 | Agricultural Systems | 20 |
| OECD | Better policies for rural development | 1996 | Better policies for rural development | 13 |
| Vidal, R.V.V. | Rural development within the EU LEADER+ programme: new tools and technologies | 2009 | AI and Society | 8 |
| Murray, M. | Planning through dialogue for rural development: The European citizens' panel initiative | 2008 | Planning Practice and Research | 5 |

### 4.2.3. Third Period 2010–2022: Maturity and New Approaches

The planning that has taken place in Europe since the 1990s until the present day, as a result of the LEADER approach, has had no methodological parallel in other parts of the world. In this third period (2010–2023), there has been significant growth in the number of articles published on the subject, showing a growing interest and broadening the intense scientific debate at an international level that emerged in the previous period. In this period, 4371 articles published in Scopus were identified, of which 808 are JCR which contain the "planning" and "rural development" keywords. The most influential references and their citations are shown in Table 6.

This period is from the second decade of the 21st century (2020), where new methodological approaches appeared, reflected in more advanced methods than those that are being implemented in the 27 European countries and which consider people as the central focus of sustainable development with adequate planning to this new path [68,69].

In this period [69], the concept of integrated rural development is renewed based on new rural governance linked to spatial planning and the development of skills, reinforcing the importance of integrating the public-private sectors, as well as mobilising local actors when it comes to sustainability.

The principles of the LEADER programme in the EU are applied to other contexts, as transnational rural development experiments [70–73], addressing new governance challenges for policy transformation. In this period, new concepts, such as resilience applied to rural communities emerge within the framework of urban–rural development relations, to achieve sustainable rural communities that are capable of surviving in the face of external factors [74]. Major topics are debated in relation to the knowledge economy, local entrepreneurship, social capital, innovation based on social learning, participatory planning [69,75,76], social structures and partnerships [71,77] for the coordination of rural development projects and policies. Other researchers [78,79], focus on analysing and understanding the changes in the rural environment in the context of the new knowledge economy.

There is continued interest in sustainable rural development planning and there is demand for new professionals who are capable of articulating bidirectional planning processes, with top-down and bottom-up models [77]. Following the Bologna Agreement, new programmes for training professionals in this field emerge within the EHEA [76,78] from both universities in the European Union and further afield. Some of these programmes combine planning models with political, social, technical, economic and environmental aspects in a novel way, for managing and evaluating projects and programmes in order to prepare professionals so that they are capable of providing integrated solutions and global challenges in international contexts with increasing urban–rural relations [79].

Through the initial stage, transition and maturity of this evolutionary process of sustainable rural development, new planning models emerge in the context of increasing urban–rural integration [80–82]. These models consider that sustainable development and rural prosperity cannot be achieved with substantial inequalities between people from rural and urban regions [83].

Amongst these new planning methods, the "Working With People" model emerges as the result of GESPLAN's 30 years of experience in the planning of sustainable rural development projects in Europe and emerging countries. The WWP model emerges as an alternative to the modern project and is the outcome of the evolving process of sustainable rural development, integrating the previous methodological approaches based on the logic of participation [42,46,84], planning as social learning [51,85], the formulation and creation of plans, and project management models that integrate behavioural competences [86–90].

Building on the conceptual foundations of Chambers' "Putting the last first", Cernea's "Putting people first" and J. Friedmann's "Planning as social learning", "Working with people" takes a new step towards connecting knowledge and action through a common project, which in addition to the technical–economic value of production (the goods and services it generates), prioritises the people that are involved in the project. The expression

"Working with people" goes beyond a "technical" vision of the project, emphasising people's behaviours in a context in which planners work together, requiring planners to have a particular social awareness and solid social ethics, in addition to technical and contextual skills [5].

The conceptualisation of the WWP model arises precisely from "Working with people", based on reality and the exchange of knowledge between people, with the researchers identifying themselves as development professionals. It responds to the very essence of rural development research, which arises from direct, face-to-face experience with people in the location where the fieldwork takes place [43].

With this approach, the WWP model, based on the theory of planning as social learning (Friedmann, 1993), is rooted in the action itself, in the form of the development project and its practical knowledge, connecting the different forms of knowledge. The WWP model involves reflection with people, knowledge and action, with the "researcher" being part of the planning team and even the project director themselves, "working with people".

Many of these experiences are published in the form of articles, so that there is a transfer of people's learning to the scientific community's knowledge system and also to the public and private agents, as well as development policy managers. These experiences, based on the three components of the WWP model (ethical–social, technical–entrepreneurial and political–contextual) have generated different methodological applications in Latin America and Europe in relation to social innovation and sustainable rural development [83,91–93], the sustainability of food production systems and rational consumption [90]; sustainable entrepreneurship [87,92,93]; the FAO principles for Responsible Investment in Agriculture (RAI principles), the Voluntary Guidelines on the Responsible Governance of Land Tenure [90,94–96] and project-based governance for sustainability [97]. These WWP applications have effectively validated the joint decision-making processes, public–private partnerships, sustainability in projects, change of mindset amongst governments and financiers, contributing to the improvement in people's quality of life. However, although local knowledge is considered essential in rural development processes [73], in many cases there is still a disconnect with action. It is therefore a significant challenge to ensure that local knowledge influences decision-making [43].

**Table 6.** Documents published and number of citations (period 2010–2022).

| Authors | Title | Year | Source Title | Cited by |
|---|---|---|---|---|
| Shucksmith, M. | Disintegrated rural development? Neo-endogenous rural development, planning and place-shaping in diffused power contexts | 2010 | Sociologia Ruralis | 241 |
| Li, Y., Westlund, H., Liu, Y. | Why some rural areas decline while some others not: An overview of rural evolution in the world | 2019 | Journal of Rural Studies | 235 |
| Neumeier, S. | Social innovation in rural development: identifying the key factors of success | 2017 | Geographical Journal | 149 |
| Long, H., Tu, S. | Rural restructuring: Theory, approach and research prospect | 2017 | Acta Geographica | 89 |
| Cazorla, A., de los Ríos, I., Salvo, M. | Working With People (WWP) in rural development projects: A proposal from social learning | 2013 | Cuadernos de Desarrollo Rural | 41 |
| Ryser, L., Halseth, G. | Rural economic development: A review of the literature from industrialised economies | 2010 | Geography Compass | 37 |
| Frank, K.I., Reiss, S.A. | The Rural Planning Perspective at an Opportune Time | 2014 | Journal of Planning Literature | 25 |

### 4.3. Analysis of the Co-Occurence of Keywords and Clustering

In this section, a keyword co-occurrence analysis is carried out, based on the associations that are established between the keywords, enabling the identification of key themes and trends in a particular area of research [17]. This analysis complements citation analysis, which has an intrinsic bias towards older studies [16].

For the purpose of the co-occurrence analysis, we firstly extracted the keywords from each of the articles selected for our dataset and analysed them with VOSViewer. Keywords that appeared at least 100 times were kept, resulting in 197 keywords that represented the main set of connected key terms. The clustering technique has been used to highlight the keyword grouping. The network of co-occurrence links between these keywords is presented through network diagrams and keyword density (Figure 3). In the network, each keyword is represented as a circle, with the size of the circle being proportional to the number of publications in which the term is found. Each colour represents a set of words grouped in a cluster, with the length of the curved lines demonstrating the approximate connection of the term's repetition and the thickness shows the strength of the relationship between the subject areas or keywords.

The results show how the researchers' contributions to rural development planning can be divided into three main clusters (Figure 5 and Table 7).

**Table 7.** Most occurrence keywords.

| Keywords and Cluster | % Links Strength | % Occurrences | % Nº Keywords |
|---|---|---|---|
| Green Cluster | 59.81% | 40.46% | 46.10% |
| Social Planning | 29.63% | 18.86% | 25.97% |
| Economic Development | 9.21% | 7.44% | 3.90% |
| Rural Development Policy | 6.91% | 4.68% | 5.84% |
| Developing Countries | 6.59% | 4.75% | 3.25% |
| Social Development | 3.88% | 2.67% | 5.19% |
| Economic factors | 3.60% | 2.07% | 1.95% |
| Red Cluster | 28.59% | 47.76% | 40.26% |
| Rural Planning | 8.56% | 16.29% | 12.99% |
| Environmental Planning | 5.38% | 5.91% | 7.79% |
| Rural Development | 3.74% | 7.16% | 3.25% |
| Urban Planning | 3.52% | 4.23% | 5.19% |
| Participatory Approach | 3.12% | 2.64% | 5.19% |
| Sustainable Development | 1.81% | 4.76% | 1.30% |
| Regional Planning | 1.41% | 4.46% | 1.30% |
| Land Use Planning | 0.93% | 2.00% | 2.60% |
| Governance Approach | 0.12% | 0.31% | 0.65% |
| Blue cluster | 11.59% | 11.78% | 13.64% |
| Organisation And Management | 3.64% | 2.91% | 4.55% |
| Human Resources | 2.52% | 2.90% | 1.95% |
| Rural Population | 1.81% | 1.64% | 0.65% |
| Project Management | 1.71% | 2.48% | 4.55% |
| Health Care Planning | 1.25% | 1.10% | 0.65% |
| Governance Approach | 0.37% | 0.34% | 0.65% |
| Community Development | 0.29% | 0.41% | 0.65% |
| Total General | 100% | 100% | 100% |

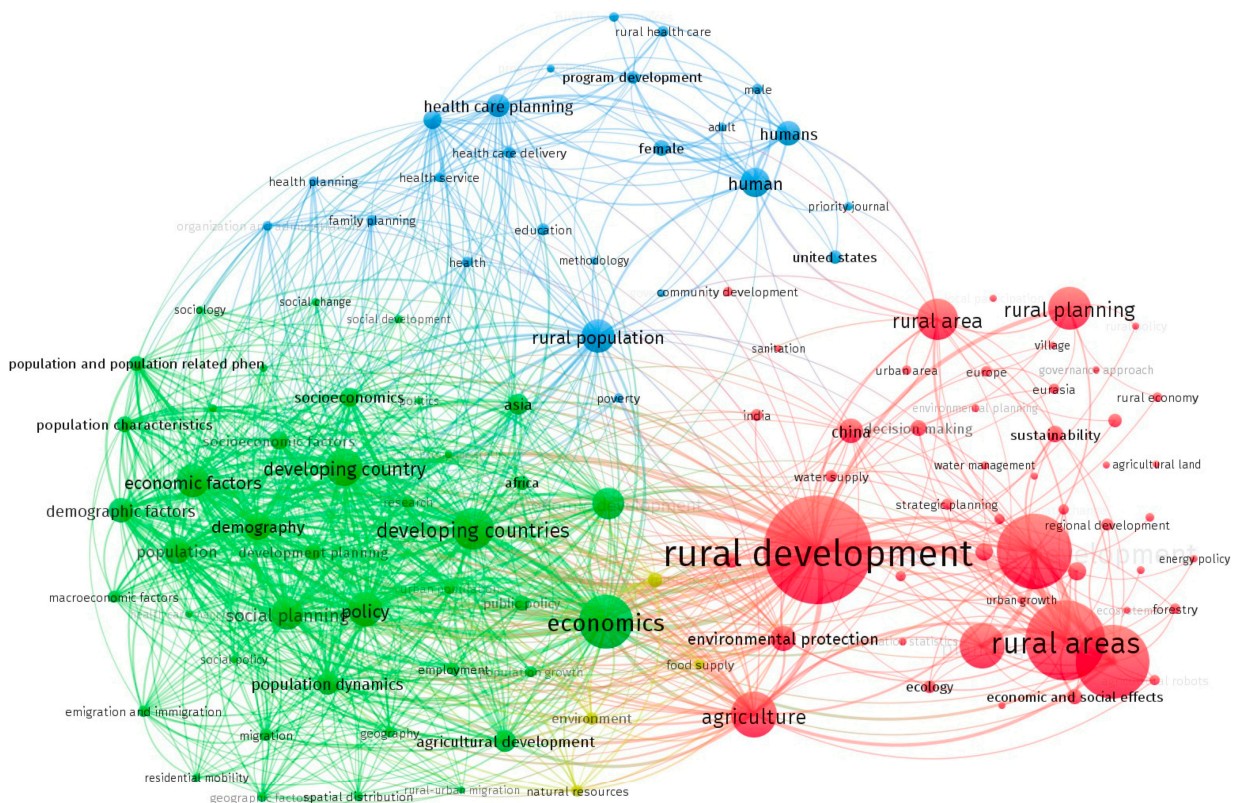

**Figure 5.** Network diagram of co-occurrence of keywords.

Red cluster: represents a group relating to sustainable rural development planning, with 40% of the keywords and 47% of co-occurrence links. In this group, the keywords rural planning, environmental planning, rural development, urban planning, participatory approach, sustainable development, regional planning and land use planning, governance approach, are particularly related through the introduction of related models and approaches. This group shows links to the social and economic aspects of the other clusters. This network is shown in the Figure 6. Robert Chambers is one of the most influential authors, with his famous "Participatory Rural Appraisal" (PRA) [42] and "Putting the first last" [43] slogan standing out. Other relevant authors, as have developed participatory methodologies applied to rural development [49,54] to overcome social conflicts in the planning of sustainable rural development using new tools and skills demanded by planners [39,54,55].

The Green cluster regarding social planning, social and economic dynamics in developing countries represents the most extensive group with 46% of the keywords and 40% of the co-occurrence links. It is the group in which the keywords rural social planning, economic development, Economic factors, developing countries, social development, are particularly related. A detail of this this cluster is shown in the Figure 7. The most relevant paper is the contribution made by Friedmann, who for the first time stood out in the concept of planning as social learning [51] and highlighted the need to provide this new paradigm in regional development [85]. Another relevant contribution is provided by Neumeier [75], who describes the social innovation in rural development, identifying the key factors of success.

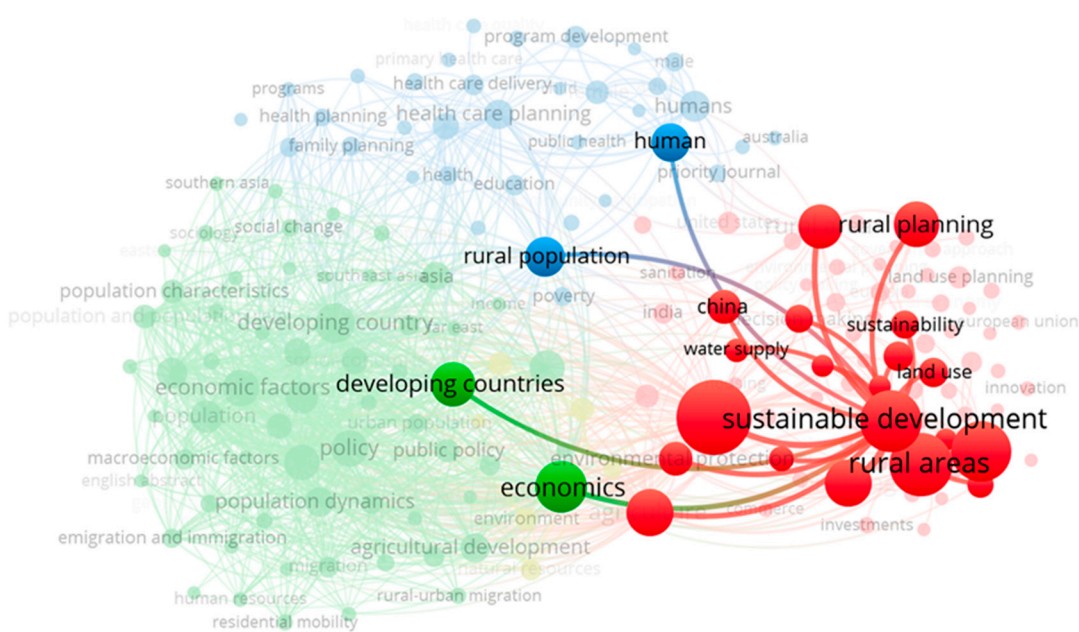

**Figure 6.** Red Cluster network diagram of co-occurrence of keywords.

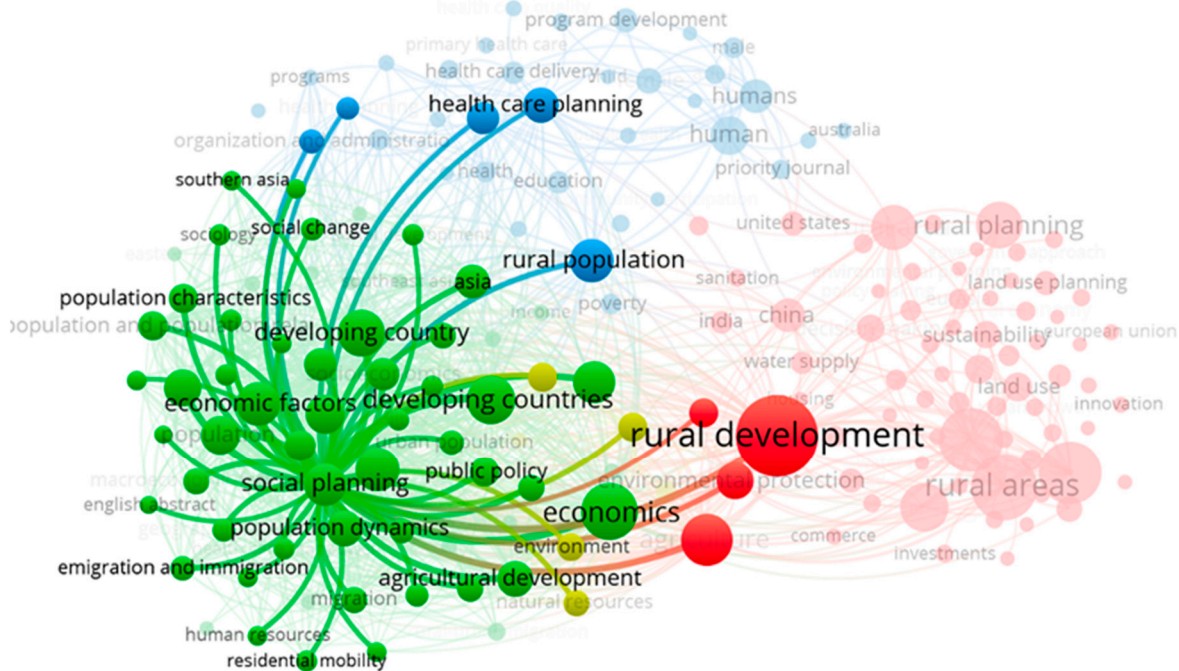

**Figure 7.** Green cluster network diagram.

Blue cluster: relating to project and programme management, governance, and the human dimension in relation to rural development. It represents the smallest of the clusters, with 13.64% of the keywords and 11.78% of the co-occurrence links. The keywords organisation and management, human resources, project management, governance approach, rural population, health care planning, community development are particularly related (Figure 8). Barke and Newton have evaluated the principles of the LEADER programme in the EU, as transnational rural development experiments, addressing new governance approach for rural development and policy transformation [57,58]. Another relevant contribution has been provided by High and Nemes who describes the new organisation and management in LEADER rural areas [59].

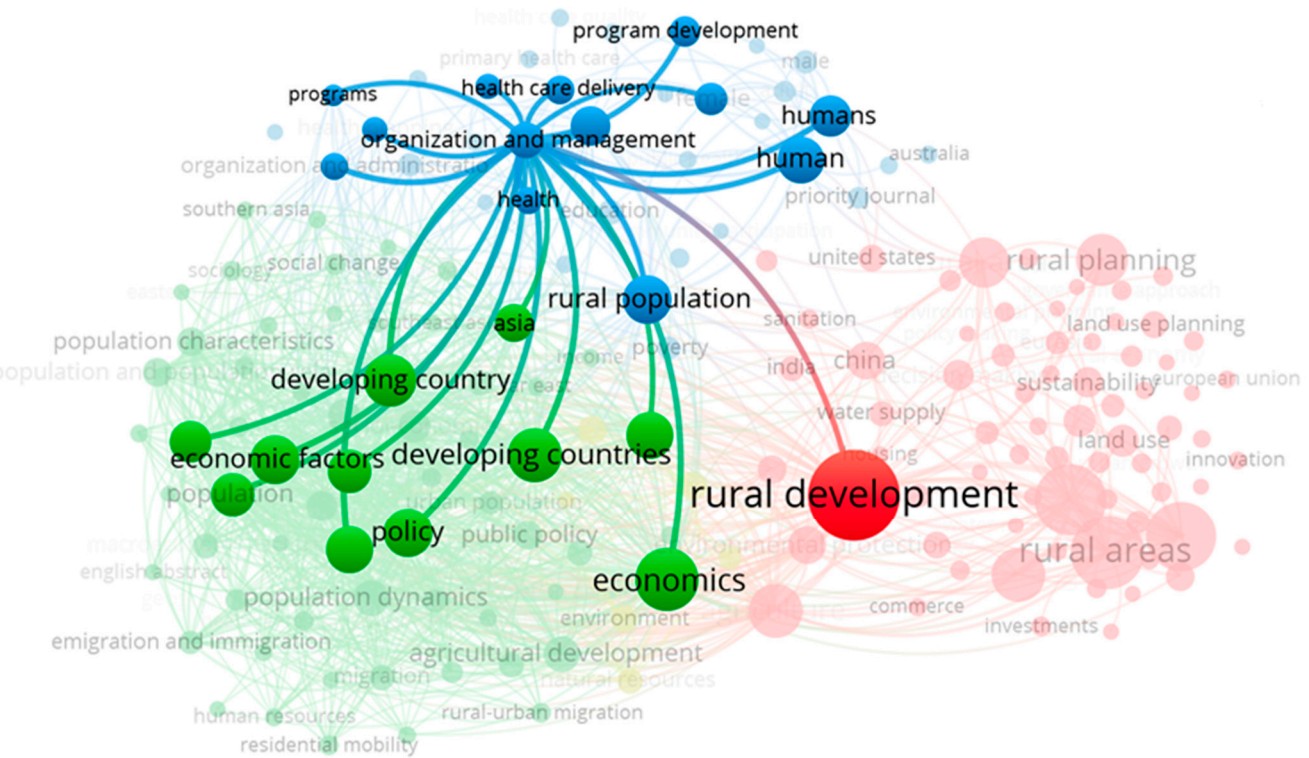

**Figure 8.** Blue cluster network diagram.

Figure 9 represents the bibliographic coupling where countries are used as a unit of analysis. We developed the international country co-network map using VOSviewer software. In Figure 9, a node represents a country, the size of the node denotes the activity of the country and a line is established when two countries have a collaborative relationship. The thickness of the line reflects the tightness of cooperation between countries. We set the threshold as 10; then there are 73 countries meeting the requirement. The VOSviewer software divides these 73 nodes into 6 clusters. One color represents one cluster.

As we can see from Figure 9, the USA, United Kingdom, China, Netherlands, Germany, Australia, Italy and Spain are the biggest nodes. The USA, Canada, Brazil, México, Colombia, Argentina England, Netherlands, Sweden, Spain and Poland belong to the blue cluster. Netherlands, Germany, Italy and Ireland belong to the red cluster. China, South Korea, Taiwan, Thailand and Australia belong to the green cluster. Therefore, geographical location is an important factor that determines international cooperation.

Table 8 presents the top 10 countries that contribute 70% of the total citations. USA has the most citations, followed by the United Kingdom, China, Netherlands and Australia.

**Table 8.** Top ten countries with the highest number of citations publications.

| Country | Documents | Citations | % Citations | Total Link Strength |
|---|---|---|---|---|
| United States | 943 | 21,202 | 16.97 | 415 |
| United Kingdom | 563 | 19,426 | 15.55 | 386 |
| China | 1180 | 14,649 | 11.72 | 307 |
| Netherlands | 178 | 5734 | 4.59 | 179 |
| Australia | 270 | 5625 | 4.50 | 174 |
| Italy | 253 | 4126 | 3.30 | 173 |
| Spain | 207 | 4074 | 3.26 | 113 |
| Germany | 209 | 3756 | 3.01 | 143 |
| India | 358 | 3714 | 2.97 | 105 |
| Canada | 190 | 3689 | 2.95 | 96 |

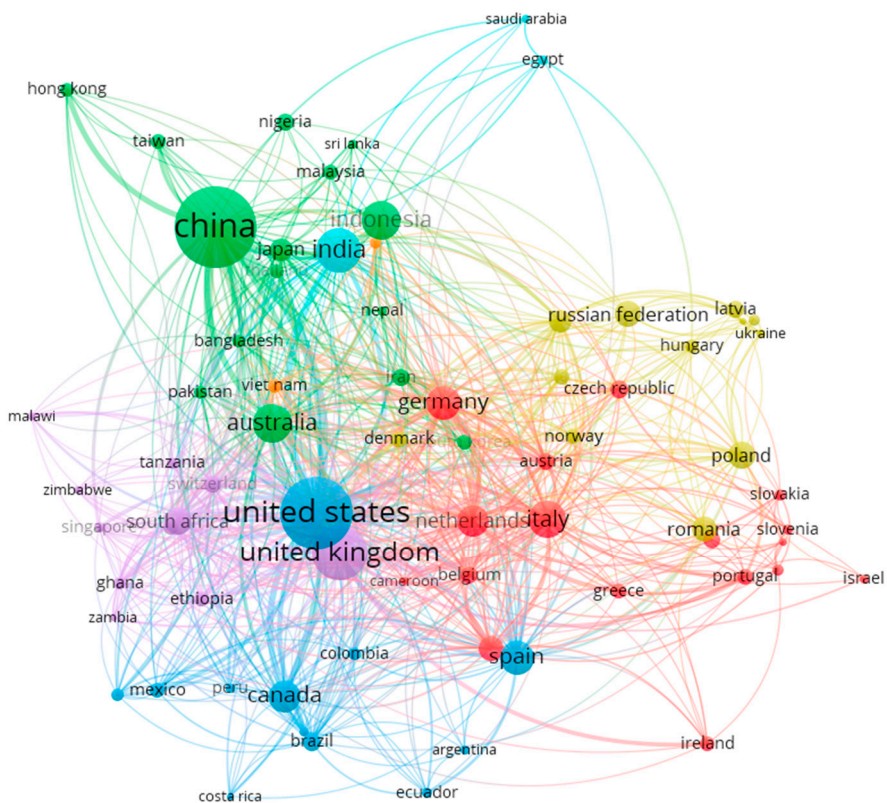

**Figure 9.** Country analysis network diagram.

### 4.4. "From Putting the Last First" to "Working with People" in Rural Development Research

Finally, this section includes a complementary analysis of the keyword co-occurrence, in relation to the three dimensions of the WWP model [5]. These three dimensions (ethical–social, technical–entrepreneurial and political–contextual) should be present in all projects designed and planned using the WWP model, with interactions and overlaps between them through the social learning processes. The results of this analysis are shown in the following Table 9, which identifies the key themes and research trends relating to the three dimensions of the WWP model. The Table 10 shows a complementary analysis of keyword co-occurrence relating the three dimensions of the WWP model with the three previous clusters.

**Table 9.** Most occurrence keywords in relation to WWP dimensions.

| WWP Dimensions and Keywords | % Total Keyword | % Total Link Strength | % Total Occurrences |
|---|---|---|---|
| **Technical—Entrepreneurial dimension** | **24.14%** | **20.17%** | **20.48%** |
| Technical dimension | 8.05% | 2.58% | 3.88% |
| Organisation and management | 4.02% | 3.41% | 2.68% |
| Project management | 4.02% | 1.60% | 2.27% |
| Economic development | 3.45% | 8.63% | 6.84% |
| Agriculture and forest | 2.30% | 3.54% | 3.77% |
| Energies | 1.15% | 0.18% | 0.45% |
| Rural development project | 0.57% | 0.13% | 0.38% |
| **Ethic—Social dimension** | **25.86%** | **32.21%** | **21.94%** |
| Community development | 0.57% | 0.28% | 0.37% |
| Human resources and personal competencies | 1.72% | 2.36% | 2.67% |
| Participatory approach | 4.60% | 2.92% | 2.43% |
| Rural population: behaviour, attitudes, values | 0.57% | 1.69% | 1.51% |
| Social development | 3.45% | 3.05% | 1.81% |
| Social learning planning | 14.94% | 21.90% | 13.15% |

**Table 9.** *Cont.*

| WWP Dimensions and Keywords | % Total Keyword | % Total Link Strength | % Total Occurrences |
|---|---|---|---|
| **Political—Contextual dimension** | **50.00%** | **47.62%** | **57.57%** |
| Developed Countries | 2.87% | 6.18% | 4.36% |
| Environmental Planning | 6.32% | 4.95% | 5.22% |
| Governance Approach | 1.15% | 0.46% | 0.60% |
| Health care Planning | 0.57% | 1.18% | 1.01% |
| Land use Planning | 2.30% | 0.87% | 1.84% |
| Regional Planning | 1.15% | 1.32% | 4.10% |
| Rural Development Planning | 2.30% | 3.38% | 6.20% |
| Rural Development Policy | 5.17% | 6.47% | 4.30% |
| Rural Planning | 11.49% | 8.02% | 14.97% |
| Social development and social support | 1.15% | 0.58% | 0.64% |
| Social Planning | 8.05% | 5.86% | 4.18% |
| Socioeconomic factors. Education, employment, safety | 1.72% | 3.37% | 1.90% |
| Sustainable Development | 1.15% | 1.70% | 4.38% |
| Urban Planning | 4.60% | 3.29% | 3.88% |
| Total general | 100.00% | 100.00% | 100.00% |

**Table 10.** Most occurrence keywords.

| WWP Dimensions | Blue Cluster | Red Cluster | Green Cluster | % Total Occurrences |
|---|---|---|---|---|
| Ethic–Social | 18.73% | 9.98% | 71.29% | 100.00% |
| Technical–Entrepreneurial | 40.51% | 44.43% | 15.06% | 100.00% |
| Political–Contextual | 2.38% | 61.75% | 35.86% | 100.00% |

Ethical–social dimension: This represents a group with 25.86% of the keywords and 22% of the co-occurrence links. This "ethical-social" cluster includes keywords relating to the behaviours, attitudes and values of the people involved in rural development projects and programmes. It also includes keywords relating to the social aspects of planning and the population's participatory processes. It is strongly identified with the green cluster (with 71.29% of the co-occurring links).

Technical–entrepreneurial dimension: This represents a group with 24.14% of the keywords with 20.48% of the co-occurrence links. This group is made up of keywords relating to the implementation and management of projects, such as investment units and "technical" instruments that generate goods and services for rural development. It also includes keywords relating to public and private economic activities (such as agriculture, tourism, etc.), and technical innovations in relation to rural development projects. This dimension overlaps with the red cluster (with 44.43% of the co-occurrence links) and blue cluster (40.5%) and to a lesser extent with the green cluster (15%).

Political–contextual dimension: This is the largest group consisting of 50% of the keywords, relating to public-administrative planning and rural development policies: rural development policy, governance approach, environmental planning, land use planning, regional planning, developing countries. This largest group is mainly focused (with 61.75% of the co-occurrence links) in the red classer, with a medium overlap in the green cluster (35.8%) and very slight overlap in the blue cluster (2.38%).

To complement this analysis, the clustering technique has also been used to highlight the influence of the "Working with people" model in the field of rural development. The Figure 10 represents the visualisation of the network that emerges from the 2860 documents in Google Scholar (GS), Scopus (Elsevier) and WoS, which include the keywords "Working with people" and "rural development" (Table 11).

**Table 11.** Search results by topic.

| Topic | Authors | Nº Documents |
|---|---|---|
| "Putting the last first" and "rural development" | Chambers, R. | 7.730 |
| "Putting people first" and "rural development" | Cernea, M.M. | 3.730 |
| "Working with people" and "rural development" | Cazorla A., De los Ríos, I. | 2.860 |
| "Planning as social learning" | Friedmann, J. | 230 |

The cloud map shows the number of occurrences of the keywords "Working with people" and "rural development". Three clusters are observed, in which "people" stand out as the central factor linking the three dimensions. This leads us to reaffirm that people are the focus of the studies relating to the WWP model as a conceptual proposal based on social learning for rural development projects. A large number of papers, put the emphasis on people, highlighting the importance of "learning from experience" [35], as a way of improving the effectiveness of sustainable rural development projects and programmes [35,47,48].

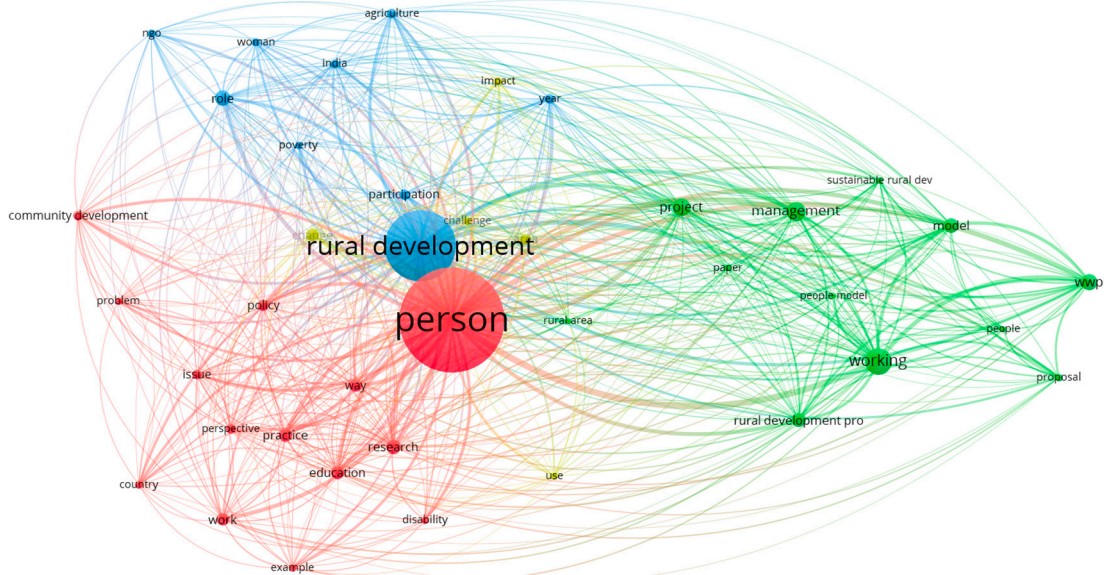

**Figure 10.** Network diagram of "Working with People and "Rural Development" keywords.

## 5. Discussion and Future Research Directions

The bibliometric analysis reveal the existence of three clusters and the shows the evolution of a new approach for the planning of sustainable rural development projects in post-modernity: Working with People (WWP). The three WWP dimensions (ethical–social, technical–entrepreneurial and political–contextual) have interactions with the three cluster and overlaps between them through the social learning processes.

This WWP has been applied in several experiences in rural development projects, especially in LEADER areas [55,56]. The analysis provides evidence that process of social learning to sustainable rural development projects can be effective for different initiatives by the public and private sectors [23,59]. Of course, this does not mean that WWP approach is always optimal in every rural context. Multiple ways and approaches can be sequenced and combined in sustainable rural development planning [61].

Analysis of the three cluster reveals explored research areas and related to WWP dimensions, such as the organisation and management, project management, environmental planning, community development, social learning planning for rural development [68], responsible governance, land use planning [95] and to innovative topics, such as the in-

troduction of new technologies for sustainable rural development and how integrating informal and formal knowledge enhances sustainable and resilient agriculture [73].

Robert Chambers considered the need for the rural development planners to take on a more humble role, listening and learning from the population and appeals to the scientific community from an ethical point of view [39], explaining the mistakes in development practice and calling for changes in learning methods, behaviours and values to prioritise people [40,41]. His call for participation in rural development has materialised in his famous "Putting the first last" [43] slogan. Cernea is another important author and, with his slogan "Putting people first", addresses the adequacy and entry points of sociological knowledge in the planning of rural development projects [46], opening up a new field of research based on the social components of sustainability. This is therefore a precursor for considering sustainability based on the three dimensions (social, economic, ecological) [47], along with other planners who incorporate the notion of care and respect based on an integrated vision [48].

Literature reviews have also suggested the planning as social learning, however, are very seldom mentioned or inscribed as components of a successful new rural development project, plan or policy from the project management competencies point of view. The evidence collected through our bibliometric analysis and literature review have revealed the social challenges faced rural development and "people" stand out as the central factor linking the three cluster.

WWP approach opens up the possibility of new research questions and new postmodern approaches to lighten existing questions in rural development projects theory and in planning as social learning research. At the core of WWP model the balance between three dimensions of competences—technical, behavioural and contextual competence— is basic, and also a balanced role of social agents of the areas of social relationships system (political, public, private and social). Future research should address the gaps regarding the three dimensions of WWP model.

First research questions are in connection with the political–contextual dimension of the rural development project. At the failure of modern project, in post modernity emerging clearly universal values and future trends, this can be extrapolated to all approaches and all circumstances. Since the Rio Conference in 2012, an increasing number of research have started to consider the Sustainable Development Goals (SDGs) and their inclusion represents an innovative pathway for academics due to the holistic character of the 2030 Agenda. Concepts such as the eradication of poverty, wellbeing, and peace have typically been analysed from a policymaking perspective rather than a scholarly one [98]. In this sense, the research to sustain the analysis of the SDGs at an organisational level will be a relevant challenge for the coming years. However, the best approach for any particular circumstance is dependent on the objectives of the intervention and the specific context. Unfortunately, most national and international development agencies assume that there is one approach (their existing policy) which is the best and they miss the essential first stage of the project cycle, not asking the question: what type of intervention approach is best suited to this type of issue in this context?

The second type of questions are related to the technical–entrepreneurial dimension of the rural development project, as an innovation unit and "technical" tool capable of generating a flow of goods and services for people. The technological innovation has dominated debates concerning development and project management and has been traditionally conceived as a simple act of production, design and engineering of product or process, without mentioning the social processes. From WWP process approach is conceiving innovation as a process of social learning [23,59] that includes new human relations, new management, administration and negotiation systems, new forms of learning, new ways of structuring and sharing information and knowledge among all social agents that bring innovation. Innovation as a process of social learning might be therefore understood as a hard, open and interactive process with an important social dimension, which means a constant adaptation of the forms of knowledge and learning to the market and technological conditions

constantly changing. WWP model integrate the planning as of social learning identified new roles for planners and the knowledge of social planning.

A third area of research question enables us to consider topics of how the behavioural competence is developed from the sustainable rural development planning project works. Beneficiary participation is essential for effective sustainable rural development interventions, but it is only one element of a systematic approach that builds on empirical experience. Social learning process required a collective dimension, interrelate different knowledge in the decision making of the actions [82]. The literature show that the multiplicity of knowledge sources and learning structures, the integrative links between informal knowledge and formal knowledge institutions demonstrated that farmers' informal knowledge makes a considerable contribution to promoting sustainable rural development and resilient agriculture [73]. The new research tendencies point towards acceleration and important changes in the ways of social learning, betting for the processes based in the action—learning by doing—as well as competence-based learning in the training of values and abilities essentially acquired through education [83]. In WWP approach, the origin of knowledge is observation and experience. This innovation as a learning process is especially important in the sustainable rural development projects, where it is demanded that the rural population change from being an object to being a subject of the projects and processes [37]; is also needed «putting the first last» [39].

Finally, and most crucial, the evidence collected through our bibliometric analysis and literature review have revealed the new challenges related to knowledge and action. Conjoint analysis of the three clusters unveils a high degree of linkages between these topics and enables us to consider questions of how knowledge—formal and informal knowledge of the population experienced with the planner's expertise—is and can be better connected to the logic of local "collective action project" [46]. Local collective action is increasingly being used to describe how civil society engages with, and acts upon, sustainability transformations. The findings emphasise that "working with people"—in several experiences in rural development projects—means navigating among different assumptions, values, and social transformation processes, which involve guiding principles as "local collective action" [99], "putting the last first" [43], "Putting people first" [46] and "planning as social learning" [23]. This evidence appears both in the literature review, with its heterogeneous results, and the keyword analysis. Several authors have encouraged the inclusion of contributions from social planning, community development and human resources research, so as to involve stakeholders and policymakers within the debate [100,101]. WWP as a learning process, starts with a perceptive activity put into practice through the view of things, thinking about them and listening to people, being always respectful with the others and from the appreciation of their values [5].

From the understanding of these questions, it will be possible to move forward to an enhancement of rural development projects, making the interventions to be more efficient and human.

## 6. Conclusions

Analysing the evolution of published articles, a significant increase can be observed to respond to the ongoing needs relating to sustainable rural development. The results of highly cited papers and journal co-citation networks demonstrate that "Social Planning" and "social learning" constitute the main topics covered by the journal nowadays. If "putting people first" is to be more than a trendy slogan, rural development planners must face the nuts and bolts of organising participation.

The evidence collected through our bibliometric analysis have revealed the new challenges in rural development and an evolution from slogan the "Putting the last first" to "Working with People". Putting people first in rural development is a recurring theme

The WWP model has been validated as a proposal in numerous contexts, integrating the social learning conceptual framework and the contributions from influential teachers, such as Chambers and Cernea. The WWP model for the "local action project" implementa-

tion seek a balance between the three dimensions (ethical–social, technical–entrepreneurial and political–contextual), from social learning and "working together", to prioritise the people. The result of this analysis identifies the main areas (cluster) and future research trends relating to the three dimensions of the WWP model.

This research suggests that WWP model going a step further in active participation (from slogan the "Putting the last first", towards "working with people") and creating joint actions that integrate experienced knowledge and expert knowledge in the formulation and management of development projects, providing mutual learning between the population and the planning team. The findings emphasise that for those engaged in rural development planning, to guarantee the social learning processes, it is necessary to have an adequate appreciation of values, to perceive other people's qualities and understand their points of view.

This leads us to affirm that the ethics and behaviours of the people involved should form the basis of the new methodological approaches, such as the WWP model, with people considered as the focal point of sustainable rural development. It is hoped that the research results will contribute to the domains related to sustainable rural development planning science, intimately linked to the human action from the people involved, engaged and working in the project.

**Author Contributions:** The authors have contributed jointly in all phases of this research. All authors have read and agreed to the published version of the manuscript.

**Funding:** This research received no external funding.

**Institutional Review Board Statement:** Not applicable.

**Informed Consent Statement:** Not applicable.

**Data Availability Statement:** Publicly available datasets were analysed in this study. This data can be found here: https://www.scopus.com, https://www.webofscience.com/wos/woscc/basic-search (accessed on 1 February 2023).

**Acknowledgments:** The authors acknowledge the technical support provided by the CSIC-Cybermetrics Lab Department (research group belonging to the Consejo Superior de Investigaciones Científicas).

**Conflicts of Interest:** The authors declare no conflict of interest.

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
