# Peer review of "From “Putting the Last First” to “Working with People” in Rural Development Planning: A Bibliometric Analysis of 50 Years of Research"

_sustainability, doi:10.3390/su151310117_

Round 1

Reviewer 1 Report

The article exhibits an intriguing topic overall. However, there is room for improvement in terms of revising and refining the general topics discussed, which would contribute to its overall strength. Additionally, the title could be enhanced to better capture the essence of the research.

In order to enhance the quality of the article, it is advisable to improve the results section by incorporating more comprehensive and in-depth analysis. It is crucial to provide a clearer explanation of the underlying processes and support the findings with relevant literature, where appropriate.

The absence of a discussion section in the article is a notable limitation. Therefore, it is recommended to include a well-developed discussion that critically examines and interprets the obtained results. This discussion should establish connections with existing knowledge and shed light on the broader implications of the findings.

One area that requires attention is the strength of the conclusions. To address this limitation, it is essential to strengthen the conclusion section by emphasizing the significance of the findings and their potential implications. This will provide a more robust and persuasive resolution to the research problem, considering the inherent interest of the topic.

Author Response

Thank you very much for the review of the paper, that helps the quality. We show you improvements made based on your comments and suggestions and those of other reviewers:

  • To improve the quality of the article, the results section has been improved incorporating more complete and in-depth analysis.
  • The concept of “Working with people” has been further explained. As suggested, a specific section has been created in the introduction.
  • We provide a clearer explanation of the underlying processes and support the findings with new literature.
  • To this end, a new and extensive discussion section has been added to the article, critically examining and interpreting the results obtained and making connections with existing knowledge in light of the findings.
  • The title has been adjusted to better capture the essence of the investigation.
  • The conclusions section has been improved, emphasizing the significance of the findings and their potential implications.

Reviewer 2 Report

·         It is suggested to explain more the concept “Working with People” (maybe in Introduction part or even like separate subchapter) – we see it in Abstract and then in 3. Results of the bibliometric analysis, and some ideas in results part. Why WWP model is suitable for this article?

·         Not very clear – what the authors wanted to find from this review/article – there were mentioned 3 clusters?

·         The numbering and visualization of tables should be revised as well placing of Figures (3, 4) should be revised.

·         Was there observed the distribution of articles based on rural development by countries – it is very interesting how this process/progress was in separate countries?

·         Conclusions could be clearer defining how the authors succeeded to use the methods, what limitations were if were, etc. The article seems to be much clear that conclusions.

Author Response

Thank you very much for the review of the paper, that helps the quality. We show you improvements made based on your comments and suggestions and those of other reviewers:

  • The concept of “Working with people” has been further explained. As suggested, a specific section has been created in the introduction.
  • It has been clarified in the results of the bibliometric analysis, and why the WWP model is suitable for this article.
  • In addition, it shows more clearly what is intended to be found in this review.
  • New tables and figures have been added. The numbering and visualization of the tables have been revised, as well as the location of the Figures.
  • A new analysis has been carried out on the distribution of articles and citations by countries.
  • The title has been adjusted to better capture the essence of the investigation.
  • The conclusions section has been improved, emphasizing the significance of the findings and their potential implications.

Round 2

Reviewer 1 Report

Dear Editor,

Upon re-reading the article, I am pleased to note that the authors have made significant improvements to the manuscript. The revisions have enhanced the overall quality of the paper, making it suitable for publication in this journal.

I recommend considering this manuscript for publication, as it aligns with the scope and objectives of the journal. The authors' efforts in addressing previous concerns and enhancing the clarity of the content are evident.